# Comparative Diagnostic Accuracy of the STANDARD M10 Assay for the Molecular Diagnosis of SARS-CoV-2 in the Point-of-Care and Critical Care Settings

**DOI:** 10.3390/jcm11092465

**Published:** 2022-04-27

**Authors:** Alexander Domnich, Andrea Orsi, Carlo-Simone Trombetta, Elisabetta Costa, Giulia Guarona, Miriana Lucente, Valentina Ricucci, Bianca Bruzzone, Giancarlo Icardi

**Affiliations:** 1Hygiene Unit, San Martino Policlinico Hospital-IRCCS for Oncology and Neurosciences, 16132 Genoa, Italy; andrea.orsi@unige.it (A.O.); giuly.guarons@outlook.it (G.G.); miriana.lucente@hsanmartino.it (M.L.); valentina.ricucci@hsanmartino.it (V.R.); bianca.bruzzone@hsanmartino.it (B.B.); icardi@unige.it (G.I.); 2Department of Health Sciences (DISSAL), University of Genoa, 16132 Genoa, Italy; s5285039@studenti.unige.it (C.-S.T.); ec240583@gmail.com (E.C.)

**Keywords:** COVID-19, SARS-CoV-2, RT-PCR, diagnostic accuracy, point-of-care testing, critical care

## Abstract

Accurate and rapid molecular diagnosis of COVID-19 is a crucial step to tackle the ongoing pandemic. The primary objective of this study was to estimate the real-world performance of the novel RT-PCR STANDARD M10 SARS-CoV-2 assay in a large number of nasopharyngeal (NP) specimens eluted in universal transport medium. The secondary objective was to evaluate the compatibility of this kit in testing NP samples eluted in an inactivated transport medium (essential for point-of-care testing) and lower respiratory tract (LRT) specimens, which are commonly collected in critical care. A total of 591 samples were analyzed. Compared with the standard extraction-based RT-PCR Allplex 2019-nCoV (time-to-result of 270 min), the sensitivities of the STANDARD M10 were 100% (95% CI: 98.1–100%), 95.5% (95% CI: 91.7–97.6%), and 99.5% (95% CI: 97.2–99.9%) for ≥1 gene, the ORF1ab gene, and the E gene, respectively, while the specificity was 100% (95% CI: 98.7–100%). The diagnostic accuracy was 100% in testing both NP samples eluted in an inactivated transport medium and LRT specimens. STANDARD M10 reliably detects SARS-CoV-2 in 60 min, may be used as a POC tool, and is suitable for testing LRT specimens in the critical care setting.

## 1. Introduction

Detection of SARS-CoV-2 viral RNA by means of real-time reverse transcription-polymerase chain reaction (RT-PCR) is considered the gold standard for the diagnosis of both symptomatic and asymptomatic COVID-19 cases [1,2]. Alongside non-specific interventions and massive immunization campaigns, the accurate diagnosis of SARS-CoV-2 is the key measure able to tackle the ongoing pandemic [3].

Standard RT-PCR assays are performed under proctored conditions and require sophisticated laboratory equipment and skilled staff. Moreover, despite some recent progress, the traditional techniques are still associated with suboptimal turnaround times [4,5]. Quick and accurate laboratory diagnosis of COVID-19 is required in some situations, such as the clinical management of critically ill patients [6]. If sufficiently accurate, faster diagnostic kits are particularly advantageous for screening programs when large numbers of samples are tested daily [7]. To address these needs, several fast point-of-care (POC) techniques have been developed and include, for example, lateral flow and fluorescent rapid antigen-detecting (RADT) and rapid nucleic acid amplification (NAAT) tests [8,9]. Although RADTs are relatively inexpensive and able to provide results in a few minutes, the use of RADTs may be associated with a high proportion of false-negative results (especially in subjects with low viral loads [8,10]), thus compromising their utility in low-incidence settings [11].

Rapid RT-PCR assays are intended to detect SARS-CoV-2 RNA in the POC and critical care settings in a reasonably short time and have become increasingly common. Indeed, a Cochrane review [8] identified a total of 32 evaluations of the diagnostic performance of five different rapid molecular assays; among these, Xpert Xpress SARS-CoV-2 (Cepheid, Sunnyvale, CA, USA) (*n* = 15) and ID Now SARS-CoV-2 (Abbott, Chicago, IL, USA) (*n* = 13) were the most frequently assessed. Compared with the standard RT-PCR assays, the pooled sensitivity and specificity of the rapid molecular tests were 95.1% (95% CI: 90.5–97.6%) and 98.8% (95% CI: 98.3–99.2%), respectively [8]. A significant between-brand variation has been also reported. For instance, the sensitivity of the Xpert Xpress kit was, on average, 19.8% (95% CI: 14.9–24.7%) higher than that of ID Now (*p* < 0.0001).

STANDARD M10 SARS-CoV-2 (SD Biosensor, Seoul, Korea) is an all-in-one cartridge multiplex qualitative RT-PCR assay currently approved for the molecular diagnosis of COVID-19 from nasopharyngeal (NP) or oropharyngeal swab specimens. The manufacturer’s declared sensitivity and specificity for these sample types are 100% [12]. However, validation studies conceived for regulatory purposes may have low external validity since they are usually conducted by testing a limited number of well-characterized samples. Indeed, no in-field evaluations of this assay have been conducted so far. The primary objective of this study was to estimate the diagnostic accuracy of the STANDARD M10 assay in a relatively large sample of routinely collected NP swab specimens. Moreover, considering that the assay is potentially relevant for rapid POC diagnosis outside centralized laboratory facilities, we aimed to validate the STANDARD M10 in detecting SARS-CoV-2 in NP samples eluted in inactivated transport media (which ensures safe sample handling and processing). Analogously, the performance of this assay in testing lower respiratory tract (LRT) samples, which are commonly used in intensive care units, was also assessed.

## 2. Materials and Methods

### 2.1. Overall Study Design and Procedures

The present study is reported according to the STARD (standards for reporting of diagnostic accuracy studies) statement [13] (Appendix A Appendix A).

The study was conducted between September 2021 and March 2022 at the regional reference laboratory for COVID-19 diagnostics, located at San Martino Policlinico Hospital in the Metropolitan City of Genoa (Italy). During the study period, most detections were due to the Delta and Omicron variants of concern (VOCs) [14].

For the main study, all samples with a conclusive result in the reference test were potentially eligible. A total of 500 (200 positive and 300 negative) NP samples swabbed by means of a sterilized cotton flock and eluted in the universal transport medium (UTM) (Copan Italia, Brescia, Italy) were tested in both reference and index tests (see below). These specimens came from both community-dwelling and hospitalized subjects. The age, sex, and sample date were readily available on the sample tube, while no clinical data could be associated with a given sample. The adopted sample size was judged sufficiently powered and determined on the basis of the Foundation for Innovative New Diagnostics (FIND) [15] and European Commission [16] recommendations; in particular, a minimum of 100 positive and 300 negative samples are recommended for clinical evaluation of rapid tests [15,16]. Indeed, considering that the false positivity rate is typically very low [17], a higher number of true negative samples is required.

All samples were analyzed in fresh and within eight hours of arrival at the laboratory. These were first tested by the reference standard RT-PCR assay available (see below). Once the results of the reference tests were obtained, the leftover samples were consecutively collected and tested in the STANDARD M10 assay. Both reference and index tests were performed on the same day.

For the complementary evaluations of the STANDARD M10 kit, we assessed its compatibility and performance in detecting SARS-CoV-2 in NP swabs eluted in the eNAT sample collection system (Copan Italia, Brescia, Italy). The eNAT system is a guanidine-thiocyanate-based medium used for virus inactivation and is particularly useful for the POC assays performed outside biosafety level 2 [18]. The eNAT samples came from patients attending the hospital emergency department for priority and rapid RT-PCR detection of SARS-CoV-2 (usually for patients who had to be urgently hospitalized). The performance of the STANDARD M10 kit was compared with other all-in-one cartridge-based RT-PCR assays available in the laboratory at that time (see below). The positivity status was further confirmed by searching the internal database for the standard RT-PCR output. To assess the compatibility of the eNAT system with the STANDARD M10 assay, a sample size of 60 (30 positive and 30 negative) eNAT samples was judged to be adequate.

Finally, we evaluated the compatibility and performance of the STANDARD M10 kit in detecting SARS-CoV-2 in LRT specimens, including broncoaspirate and bronchoalveolar lavage samples. These sample types came from the hospital intensive care units and were collected in sterile plastic tubes. Owing to their particular viscosity characteristics, LRT samples were first pre-treated with a liquifying dithiothreitol-based solution (Sputasol Liquid, Oxoid Limited, Basingstoke, UK) at a 1:4 ratio, vortexed, and left to stand for 15 min.

### 2.2. Index Test

The STANDARD M10 assay is an all-in-one cartridge-based ready-to-use multiplex (FAM, HEX, and Cy5 channels) RT-PCR kit intended for the qualitative detection of SARS-CoV-2 RNA. RT-PCR is performed on the STANDARD M10 analyzer (SD Biosensor, Seoul, Korea) with up to 8 scalable modules. The assay targets the ORF1ab and E genes, and the resulting amplification curves are displayed for both gene targets. Detection of the ORF1ab gene independently from the E gene precludes a “Positive” result, while samples for which only the E gene is detected are deemed “Presumptive positive”. “Invalid” results are displayed when the internal control was not amplified or target signals did not have a Ct within the valid range [12]. Providing a sufficient sample volume, all samples with invalid results were repeated. For this reason, there was no possibility to repeat tests on 1 mL eNAT samples.

Testing on the STANDARD M10 was performed according to the manufacturer’s instructions [12]. Briefly, following a short period of vortexing, a sample input volume of 600 µL was added directly to the cartridge and set up for RT-PCR. The hands-on sample preparation time is approximately 1 min, while the time-to-result is 60 min. An early call option is also available for positive samples. The index assay is currently approved for the molecular diagnosis of COVID-19 from NP or oropharyngeal swab specimens, and according to the manufacturer, the limit of detection (LOD) is 100 copies/mL [12].

Finally, the LOD for the eNAT samples was determined by using the AccuPlex SARS-CoV-2 reference kit (SeraCare, Milford, MA, USA), which contains positive reference materials for the whole genome at a concentration of 5000 copies/mL. A total of six serial dilutions (600, 300, 150, 75, 37.5, and 18.75 copies/mL) were prepared using the eNAT liquid and the positive control. Each dilution was tested in six replicates.

### 2.3. Reference Tests

Reference RT-PCR assays used in the present study were those available in the laboratory for routine diagnostics. For the main study of testing NP swabs eluted in UTM, the standard extraction-based qualitative RT-PCR Allplex 2019-nCoV (Seegene, Seoul, Korea) was used. The whole procedure followed the instructions provided by the manufacturer [19]. Briefly, the total RNA was first extracted by means of the STARMag Universal Cartridge Kit (Seegene, Seoul, Korea) on the automated Nimbus IVD (Seegene, Seoul, Korea) system. For this purpose, 200 µL of each sample was extracted and eluted with 100 µL of elution buffer and set up for RT-PCR. RT-PCR was then performed on a CFX96 instrument (Bio-Rad Laboratories, Hercules, CA, USA) with Allplex 2019-nCoV assay. The latter is a multiplex RT-PCR assay that simultaneously detects three different genes targeting the nucleoprotein (N), RNA-dependent RNA-polymerase (RdRp), and envelope (E) regions. Amplification was performed at the following temperature regimens: 50 °C for 20 min followed by 95 °C for 15 min, 45 cycles at 95 °C for 10 s, and 60 °C for 15 s for first acquisition and 72 °C for 10 s for second acquisition on the CFX96 thermal cycler. For each RT-PCR, a total of 5 µL of the extracted RNA in a final volume of 20 µL was used. Amplificons were then tested by FAM (E gene), HEX (internal control), Cal Red 610 (RdRP), and Quasar 670 (N) fluorophores. The amplification curves were finally read using the 2019-nCoV viewer (Seegene, Seoul, Korea), and samples showing a cycle threshold (Ct) value of <40 for at least two genes were deemed true positives. According to the manufacturer, the LODs are 4167, 1250, and 4167 copies/mL for E, RdRp, and N genes, respectively, while the analytical specificity is 100% [19]. The average time-to-result of this method is 270 min [4].

The Alinity m Resp-4-Plex kit (Abbott, Abbott Park, IL, USA) was used as a reference test for LRT specimens and as a resolver test for NP swabs eluted in UTM and eNAT. This assay is a multiplex RT-PCR for use with the automated Alinity m system (Abbott, Abbott Park, IL, USA) for the qualitative detection and differentiation of RNA from SARS-CoV-2, influenza A and B, and respiratory syncytial viruses. Concerning SARS-CoV-2, the RdRp and N genes are targeted, and positive results are displayed as a unique Ct value. According to the manufacturer, the LOD is 30 genome equivalent units/mL, and the time to the first result is <115 min [20].

Finally, for NP swabs eluted in the eNAT system, two different all-in-one cartridge-based rapid RT-PCR assays were used, namely Vivalytic SARS-CoV-2 (Bosch Healthcare Solutions, Waiblingen, Germany) [21] and Novodiag COVID-19 (Mobidiag, a Hologic Company, Espoo, Finland) [22]. This choice was determined by the change in the hospital procurement for rapid RT-PCR tests. In particular, samples collected before 31 December 2021 were tested in the Vivalytic SARS-CoV-2 assay, while samples collected on and after 1 January 2022 were processed using the Novodiag COVID-19 kit. The former kit targets the E gene only, while the latter targets the E and N genes. According to the manufacturers, the LODs are 750 and 313 copies/mL for the Vivalytic SARS-CoV-2 [21] and Novodiag COVID-19 [22] assays, respectively. The times to obtain results are 39 and 80 min for Vivalytic SARS-CoV-2 and Novodiag COVID-19 kits, respectively.

### 2.4. Data Analysis

Categorical variables are expressed as proportions with 95% CIs, while continuous variables are expressed as means ± standard deviations or medians with interquartile ranges (IQRs). No imputation was performed since no missing data occurred. Pearson’s *r* coefficient was computed to establish a correlation between Ct values for the E gene (being the only common target) provided by the index and reference tests. Paired *t*-tests were used to compare average E gene Ct values. The relative diagnostic accuracy of the index vs. reference standard tests was quantified by the overall accuracy, sensitivity, specificity, and Cohen’s *κ*. Hypothetical positive (PPV) and negative (NPV) predictive value curves were constructed from the observed sensitivity and specificity and by varying the disease incidence from 0 to 50%. Probit regression was used to estimate the 95% LOD for the eNAT samples.

Statistical analysis was carried out using R stat packages v. 4.1.0 (R Core Team, Vienna, Austria).

## 3. Results

A total of 200 positive and 300 negative NP swabs eluted in UTM were analyzed in both Allplex 2019-nCoV and STANDARD M10 assays (Appendix A Appendix A). The median age of the subjects was 58 (IQR: 38–76) years, and both sexes were equally distributed (49.8% of females). Samples that tested positive in the Allplex 2019-nCoV reference assay had a wide range of viral loads, with average Ct values of 24.6 ± 6.6 (range: 14–39), 25.6 ± 6.0 (range: 15–40), and 24.0 ± 6.3 (range: 13–38) for the N, RdRp, and E genes, respectively. In the Allplex 2019-nCoV assay, the prevalence of target gene dropouts were 0.5, 10.0, and 3.0% for the N, RdRp, and E genes, respectively.

A total of 9 of 500 samples (1.8%; 95% CI: 0.8–3.4%) produced invalid results in the first STANDARD M10 run and were repeated. The second run was successful for all nine samples. Table 1 reports the raw data on the detections. Briefly, the detection rates of the E and ORF1ab genes in the STANDARD M10 assay were 99.5 and 95.5%, respectively. Regarding the automatic result interpretation, 95.5% (191/200) and 4.5% (9/200) of samples were displayed as “Positive” (detection of both targets or detection of the ORF1ab gene only) and “Presumptive positive” (detection of the E gene only), respectively. As shown in Table 2, in the reference standard assay, most of these samples had RdRp dropout and Ct values of ≥35.

The overall relative accuracies of the STANDARD M10 assay were 99.8 and 98.2% for the E and ORF1ab genes, respectively. The corresponding sensitivity parameters were 99.5 and 95.5%, respectively. When considering the detection of at least one gene target, the between-assay agreement was perfect, with a relative sensitivity of the STANDARD M10 of 100%. No false-positive results emerged, giving a specificity of 100% (Table 3).

As the specificity was 100%, the PPV was 100% independent of the SARS-CoV-2 positivity prevalence. As shown in Figure 1, by varying the hypothetical disease prevalence from 0% to 50%, the NPVs for the ORF1ab and E genes were constantly above 95 and 99%, respectively.

The Ct values for the E gene in the Allplex 2019-nCoV assay (mean 24.0 ± 6.3) were, on average, 2.2 (95% CI: 2.0–2.5) points higher (*p* < 0.001) than those determined by the STANDARD M10 kit (mean 21.8 ± 6.1), confirming the lower LOD of the latter. The E gene Ct values determined by both assays were highly correlated (*p* < 0.001), with a Pearson’s *r* of 0.954 (95% CI: 0.939–0.965) (Appendix A Appendix A).

For the first complementary study, a total of 30 positive and 30 negative eNAT NP samples were tested in both STANDARD M10 and reference tests. All positive patients were also positive in the standard Allplex 2019-nCoV assay performed on samples collected in the UTM system. The reference test for the first 23 samples was the Vivalytic SARS-CoV-2 assay, while the remaining 37 samples were processed using the Novodiag COVID-19 kit. The agreement between the STANDARD M10 and Vivalytic SARS-CoV-2 was perfect (100%). By contrast, concerning the comparison between the STANDARD M10 and Novodiag COVID-19, there was one discordant result. In particular, this eNAT sample was negative for both gene targets in the Novodiag COVID-19 assay but positive for both the ORF1ab (Ct 33) and E (Ct 31) targets in the STANDARD M10 kit. This sample was therefore tested in the Alinity m Resp-4-Plex assay, and the result was positive (Ct 32). The rate of invalid results was identical (5.0%, 3/60) for both index and reference tests; because of the limited sample volume, these eNAT samples could not be retested. The overall accuracy, sensitivity, and specificity of the STANDARD M10 assay were thus 100% (95% CI: 93.7–100%), 100% (95% CI: 87.5–100%), and 100% (95% CI: 88.7–100%), respectively. The estimated LOD to achieve a 95% detection rate for NP swabs eluted in eNAT was 235 copies/mL (Appendix A Appendix A).

Finally, a total of 31 LRT samples were tested. Of these, 9 and 22 were determined as positive and negative in the Alinity m Resp-4-Plex assay, respectively. In the first STANDARD M10 run, there was one invalid result (3.2%), which turned positive in the second run. In the STANDARD M10 assay, the ORF1ab and E genes were detected in 88.9% (8/9) and 100% (9/9) of samples, respectively. The corresponding sensitivity parameters for ≥1 target, the ORF1ab gene, and the E gene were 100% (95% CI: 70.1–100%), 88.9% (95% CI: 56.5–98.0%), and 100% (95% CI: 70.1–100%), respectively. The only “Presumptive positive” results had Alinity m Resp-4-Plex Ct values of 35. No false-negative results were observed (specificity 100%; 95% CI: 85.1–100%).

## 4. Discussion

This study is the first in-field study to evaluate the diagnostic performance of the novel RT-PCR kit STANDARD M10 for the rapid molecular diagnosis of SARS-CoV-2 in NP specimens with a wide range of viral loads. Generating real-world evidence on the performance of diagnostic tools for SARS-CoV-2 has become increasingly important [23]. The observed diagnostic accuracy parameters were in line with those declared by the manufacturer [12]. The performance of the STANDARD M10 kit was fully comparable to that of the widely used Xpert Xpress kit [8]. Furthermore, we demonstrated the consistency of the STANDARD M10 assay in detecting viral RNA from NP collected in alternative inactivating collection kits and LRT specimens.

In the main study, the sensitivity for at least 1 gene target and the specificity were both 100%. On the other hand, the estimated sensitivity of the STANDARD M10 was higher for the E gene than for the ORF1ab gene; indeed, the latter was not detected in 4.5% of positive samples. A significant proportion of the ORF1ab dropout has been reported for both Alpha [24] and Delta [25] VOCs. Notably, among Alpha VOCs detected in Lebanon, the ORF1ab dropout was observed in all samples with S gene dropout [24]. Similarly, in our study, 6 of 9 samples negative for the ORF1ab gene had RdRp gene dropout in the Allplex 2019-nCoV assay. Although the ORF1ab gene is highly specific, it is considered to be less sensitive than other targets [26]. By contrast, among the possible RT-PCR targets, the E gene is the least specific, as it shows substantial sequence homology to other seasonal coronaviruses [27]. Therefore, we believe that in low-incidence settings (especially outside the winter season when circulation of other coronaviruses is uncommon), the detection of the E gene alone is suggestive of SARS-CoV-2 infection. Otherwise, these “presumptive positive” samples may be retested using an alternative assay with different gene targets.

The handling and processing of specimens present a certain level of biohazard. In this regard, immediate viral inactivation prior to sample processing is an ideal solution for the POC setting since chemical inactivation may eliminate any risk of aerosol or droplet generation [28,29]. In this study, we showed that the STANDARD M10 SARS-CoV-2 kit is fully compatible with guanidine-based inactivating swab collection kits, which makes the assay extremely useful for POC or near-POC COVID-19 diagnosis. Similar results were reported in some previous studies evaluating the eNAT samples in alternative rapid RT-PCR assays, such as Xpert Xpress [18,28,29].

We then demonstrated that the STANDARD M10 assay may be also used for the efficient detection of SARS-CoV-2 in LRT specimens. Critically ill patients are often under invasive mechanical ventilation, and the collection of upper respiratory tract samples is usually not possible under these circumstances. Moreover, testing LRT samples may be useful in patients under clinical investigation whose NP swabs are negative [30]. To our knowledge, only a few studies [6,30,31,32,33] have evaluated the diagnostic accuracy of the commercially available rapid RT-PCR kits for detecting SARS-CoV-2 RNA in these types of specimens. The protocol adopted for the pretreatment of LRT samples was judged to be efficient from the point of view of both sample handling time and test performance. LRT specimens (especially bronchoalveolar liquid) may present different densities/viscosities, and therefore, the volume of a liquifying solution may be different. In the present study, however, we decided to make the protocol uniform since, from our experience, a dilution of 1:4 is efficient for both low- and high-density LRT samples.

Finally, the observed “invalid” result rate was relatively low (1.8%). In our opinion, the rate of invalid results should always be recorded in validation studies as well as product-related materials. Indeed, high invalid result rates may compromise both the speed and efficiency of laboratory diagnostics (e.g., the necessity to repeat the RT-PCR run for urgent samples or even repeat a swab in the case of an insufficient sample volume) and the product cost-effectiveness profile.

The main limitation of the present study is that we were not able to link the RT-PCR output to patients’ clinical characteristics, such as the presence or absence of symptoms or days since the onset of symptoms. Indeed, our evaluation was conducted under real-world conditions, and such information is often unknown to the laboratory personnel. However, we believe that this shortcoming has a limited impact on the conclusions since the STANDARD M10 index test results were highly congruent with those obtained by the routinely used standard RT-PCR assay targeting three genes. Second, the two complementary studies were designed as feasibility evaluations of the compatibility of the index test with alternative collection kits and specimen types and are thus not sufficiently powered to estimate the false-positive and false-negative rates. Larger in-field evaluations of the STANDARD M10 assay on NP swab samples eluted in the eNAT or similar inactivating systems and LRT specimens are warranted. Third, the present study included only samples positive for the two most recent Delta and Omicron VOCs, and therefore, our findings may be not generalizable to both previously circulating and future viral populations. Indeed, considering a continuous evolution of SARS-CoV-2 [34], the performance of the assay should be constantly monitored, and any relevant issues (e.g., unusual gene-specific amplification curve patterns or gene dropouts) should be promptly notified to both the manufacturer and the regulators.

In conclusion, the STANDARD M10 assay is a reliable kit for the rapid molecular diagnosis of SARS-CoV-2 infection, may be used as a POC tool, and is suitable for testing LRT specimens in the critical care setting.

## Figures and Tables

**Figure 1 jcm-11-02465-f001:**
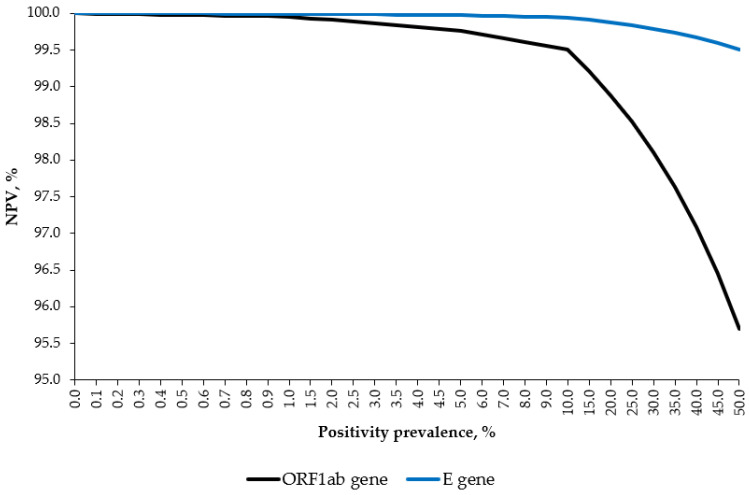
Negative predictive value (NPV) curves of the STANDARD M10 assay, by hypothetical prevalence and gene target.

**Table 1 jcm-11-02465-t001:** Raw data on target detections by gene target and assay (*n* = 500).

Sample Type	Gene Target	Allplex 2019-nCoV Assay, % (*n*)		STANDARD M10 Assay, % (*n*)	
		Detected	Not Detected	Detected	Not Detected
Positive	N	99.5 (199)	0.5 (1)	–	–
	RdRp	90.0 (180)	10.0 (20)	–	–
	E	97.0 (194)	3.0 (6)	99.5 (199)	0.5 (1)
	ORF1ab	–	–	95.5 (191)	4.5 (9)
	≥1	100 (200)	0 (0)	100 (200)	0 (0)
Negative	N	0 (0)	100 (300)	–	–
	RdRp	0 (0)	100 (300)	–	–
	E	0 (0)	100 (300)	0 (0)	100 (300)
	ORF1ab	–	–	0 (0)	100 (300)
	≥1	0 (0)	100 (300)	0 (0)	100 (300)

**Table 2 jcm-11-02465-t002:** Characteristics of samples with “Presumptive positive” result label in the STANDARD M10 assay (*n* = 9).

Sample	Ct Values in the Allplex 2019-nCoV Assay			Ct Values in the STANDARD M10 Assay	
	N	RdRp	E	ORF1ab	E
1	27	29	27	ND	33
2	32	34	31	ND	30
3	38	ND	38	ND	32
4	37	ND	38	ND	32
5	39	ND	36	ND	29
6	29	ND	30	ND	28
7	33	ND	34	ND	36
8	37	ND	35	ND	33
9	34	40	32	ND	21

ND—not detected.

**Table 3 jcm-11-02465-t003:** Relative diagnostic accuracy of the STANDARD M10 assay by parameter and gene target (*n* = 500).

Parameter	ORF1ab Gene	E Gene	≥1 Gene
Accuracy, % (95% CI)	98.2 (96.6–99.1)	99.8 (98.9–100)	100 (99.2–100)
Sensitivity, % (95% CI)	95.5 (91.7–97.6)	99.5 (97.2–99.9)	100 (98.1–100)
Specificity, % (95% CI)	100 (98.7–100)	100 (98.7–100)	100 (98.7–100)
Cohen’s *κ* (95% CI)	0.962 (0.875–1)	0.996 (0.908–1)	1 (0.912–1)

## Data Availability

All relevant data are available within the manuscript. Further details may be obtained from the corresponding author upon reasonable request and prior permission of the study funder.

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
