# Peer review of "Comparative Diagnostic Accuracy of the STANDARD M10 Assay for the Molecular Diagnosis of SARS-CoV-2 in the Point-of-Care and Critical Care Settings"

_jcm, 2022, doi:10.3390/jcm11092465_

Round 1

Reviewer 1 Report

I appreciate about a good opportunity to Review your new paper:

Comparative Diagnostic Accuracy of the STANDARD M10 Assay for the
Molecular Diagnosis of SARS-CoV-2 in the Point-of-Care and Critical Care
Settings for the  Journal of Clinical Medicine.

Overall it is very interesting and well- writing paper.

There was also interesting publication in last year in Journal of Clinical medicine that I suggest should be included in Discussion part and references : by Dierks S et al, J Clin Med 2021 10 (11): 2404

Author Response

Comment: I appreciate about a good opportunity to Review your new paper: Comparative Diagnostic Accuracy of the STANDARD M10 Assay for the Molecular Diagnosis of SARS-CoV-2 in the Point-of-Care and Critical Care Settings for the  Journal of Clinical Medicine. Overall it is very interesting and well- writing paper.

Reply: Thank you for your interest in our paper. Your comment has been addressed.

Comment: There was also interesting publication in last year in Journal of Clinical medicine that I suggest should be included in Discussion part and references: by Dierks S et al, J Clin Med 2021 10 (11): 2404

Reply: As suggested, the paper by Dierks et al. has been added to the Discussion.

Reviewer 2 Report

Thank you for the paper

I still need clarification of the following

  • Sample size (200 positive vs 300 ) negative were not justified. Why not considering the same number of samples for comparison?
  • Its not clear in the methodology wither the same sample tested for the reference test were used for index test or another samples were collected . This sentence is ambiguous "On the same day, a convenience sample of NP specimens with known result were 
    consecutively collected and tested in the STANDARD M10 assay"
  • I suggest using two graphs (bars, pie charts) to describe fig 1 than flow chart

Good luck

Author Response

Comment: Thank you for the paper. I still need clarification of the following.

Reply: Thank you for your interest in our paper. All your comment has been addressed.

Comment: Sample size (200 positive vs 300) negative were not justified. Why not considering the same number of samples for comparison?

Reply: According to the European Commission Directorate-General for Health the “target population considered in the context of an independent validation study should be based on at least 100 RT-PCR positive samples and at least 300 RT-PCR negative samples”. Indeed, considering that the false positivity rate is typically very low, a higher number of true negative samples is required. Appropriate text modifications have been made.

Comment: Its not clear in the methodology wither the same sample tested for the reference test were used for index test or another samples were collected. This sentence is ambiguous "On the same day, a convenience sample of NP specimens with known result were consecutively collected and tested in the STANDARD M10 assay"

Reply: As suggested, the phrase has been now clarified.

Comment: I suggest using two graphs (bars, pie charts) to describe fig 1 than flow chart

Reply: Thank As we reported in the Methods, the paper was reported according to the STARD (standards for reporting of diagnostic accuracy studies) statement, which “helps peer reviewers, editors and other readers in verifying that submitted and published manuscripts of diagnostic accuracy studies are sufficiently detailed” (Cohen et al. doi: 10.1136/bmjopen-2016-012799). The flowchart reported in Figure S1 is an essential part of the STARD checklist and should not be modified.